# The pro-inflammatory response to influenza A virus infection is fueled by endothelial cells

Lisa Bauer*, Laurine C Rijsbergen*, Lonneke Leijten, Feline FW Benavides, Danny Noack, Mart M Lamers, Bart L Haagmans, Rory D de Vries, Rik L de Swart, Debby van Riel

**Morbidity and mortality from influenza are associated with high levels of systemic inflammation. Endothelial cells play a key role in systemic inflammatory responses during severe influenza A virus (IAV) infections, despite being rarely infected in humans. How endothelial cells contribute to systemic inflammatory responses is unclear. Here, we developed a transwell system in which airway organoid–derived differentiated human lung epithelial cells were co-cultured with primary human lung microvascular endothelial cells (LMECs). We compared the susceptibility of LMECs to pandemic H1N1 virus and recent seasonal H1N1 and H3N2 viruses and assessed the associated pro-inflammatory responses. Despite the detection of IAV nucleoprotein in LMEC mono-cultures, there was no evidence for productive infection. In epithelial–endothelial co-cultures, abundant IAV infection of epithelial cells resulted in the breakdown of the epithelial barrier, but infection of LMECs was rarely detected. We observed a significantly higher secretion of pro-inflammatory cytokines in LMECs when co-cultured with IAV-infected epithelial cells than LMEC mono-cultures exposed to IAV. Taken together, our data show that LMECs are abortively infected by IAV but can fuel the inflammatory response.**

## Introduction

Influenza A virus (IAV) infections often cause mild, self-limiting respiratory disease but can also result in severe disease with high morbidity and mortality. Severe disease occurs especially in risk groups such as young infants, pregnant women, patients with comorbidities, and the geriatric population [1, 2]. The severity of IAV infections is dependent on many factors such as virus replication and the hosts' immune response [1, 3, 4]. For example, a dysregulation of the pro-inflammatory cytokine response in the lung and high levels of pro-inflammatory cytokines in the blood of infected individuals are associated with high morbidity caused by pandemic

H1N1 viruses, either from 1918 or 2009 or zoonotic IAV infections [5, 6, 7]. This inflammatory response is often referred to as a cytokine storm [7, 8, 9].

Endothelial cells are involved in the early cytokine and chemokine response and in the recruitment of innate immune cells to the lung during IAV infection in mammals. Although early immune responses are essential to control virus infection, exaggerated or dysregulated cytokine and chemokine responses can contribute to the pathogenesis of IAV infections. It has been shown that suppression of early innate immune responses in endothelial cells reduces the levels of systemic cytokines and decreases tissue damage, both mechanisms contributing to a greatly reduced mortality of experimentally infected animals [10]. However, how endothelial cells contribute to the systemic immune responses is unclear. IAV antigen is rarely detected in human endothelial cells in vivo after infection with zoonotic, pandemic, or seasonal IAV in humans or animal models [10, 11, 12]. Several in vitro studies show that highly pathogenic IAV, thus viruses with multibasic cleavage sites such as H5N1 and H7N9, productively replicate in human endothelial cells [13, 14, 15, 16, 17]. In contrast, seasonal and pandemic IAV without multibasic cleavage sites such as H1N1, H2N2, and H3N2 viruses cause abortive infections of endothelial cells but induce a wide array for pro-inflammatory cytokines [10, 11]. Therefore, it is hypothesized that seasonal or pandemic IAV might augment a pro-inflammatory response in endothelial cells through intracellular pathogen recognition receptors (PRRs) sensing viral RNA [18, 19, 20, 21]. Cytokines and chemokines produced by endothelial cells would directly be released into the bloodstream and contribute to the systemic inflammatory response [20, 21].

To investigate the role of endothelial cells to the pro-inflammatory response during IAV infection, we developed an in vitro transwell model in which epithelial cells (airway organoids cultured at air–liquid interface [AO at ALI]) and endothelial cells (primary lung microvascular endothelial cells [LMECs]) are co-cultured, to mimic the in vivo situation in the lower respiratory tract. This model allowed us to investigate the contribution of endothelial cells to pro-inflammatory cytokine production after infection of epithelial cells with pandemic or seasonal IAV.

Department of Viroscience, Erasmus MC, Rotterdam, The Netherlands

Correspondence: l.bauer@erasmusmc.nl; d.vanriel@erasmusmc.nl
*Lisa Bauer and Laurine C Rijsbergen contributed equally to this work

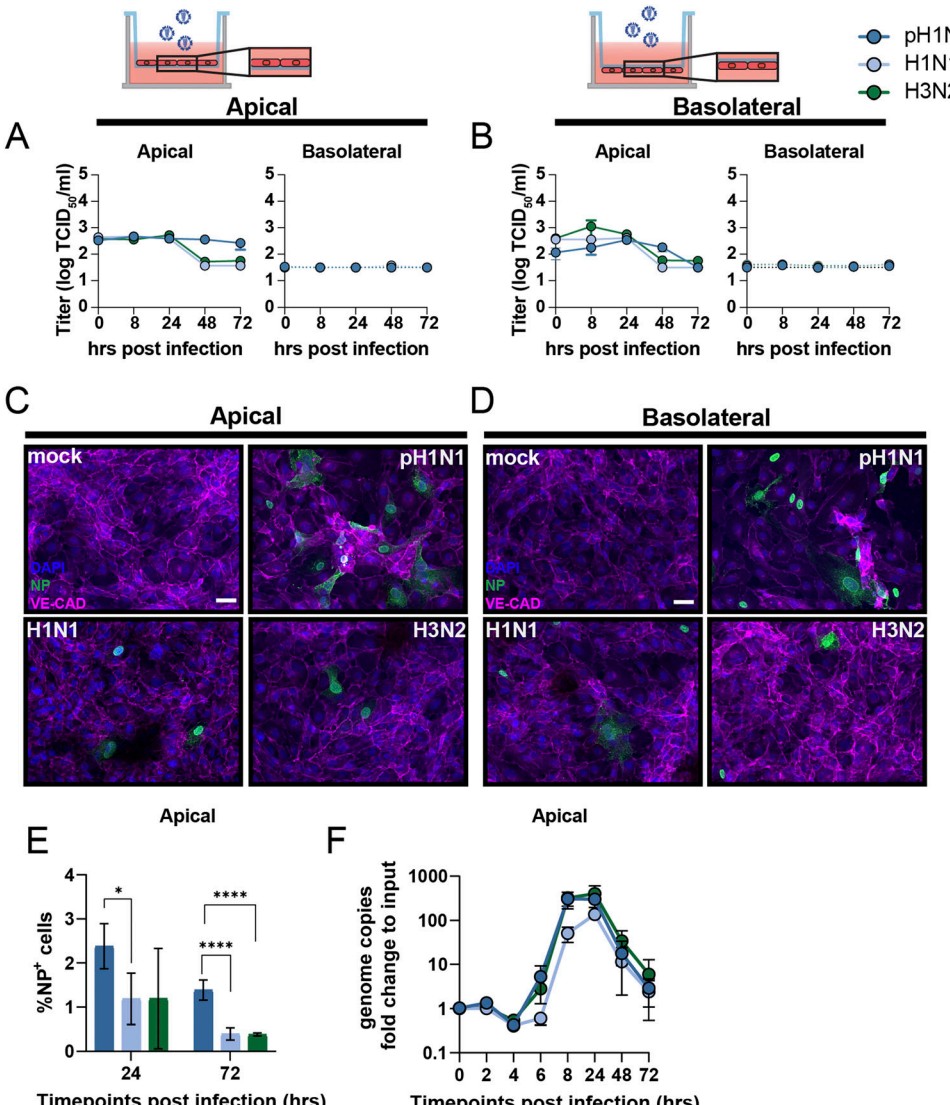

**Figure 1. Abortive infection of influenza A virus in lung microvascular endothelial cells (LMECs).**
**(A, B)** To evaluate replication efficiency, LMECs were plated on the (A) apical or (B) basolateral side of a transwell filter. LMECs were inoculated with pH1N1, H1N1, or H3N2 virus at MOI 1, and at the indicated timepoints supernatants of the apical and basolateral sides were harvested, and virus titers were determined by endpoint titration. Infection efficiency was determined by immunofluorescence staining. **(C, D)** LMECs plated on the (C) apical and (D) basolateral sides were inoculated with pH1N1, H1N1, or H3N2 virus. Cells were fixed 24 h post-inoculation and stained for the endothelial cell marker vascular endothelial-cadherin (VE-CAD, magenta) and influenza A virus nucleoprotein (NP, green). Hoechst (blue) was used to visualize nuclei. **(E)** Percentage of infection determined by flow cytometry at 24 and 72 h post-inoculation. **(F)** Intracellular viral RNA genome copies were quantified by quantitative real-time PCR at indicated timepoints. Data represent mean ± SD from at least three independent experiments performed in biological duplicates, and flow cytometry was performed in biological triplicates. A one-way ANOVA multiple comparison test was used to compare groups (*< 0.05, **<0.01, ***<0.005). Scale bar: 20 $\mu$m.

# Results

## Inefficient replication of influenza A virus in LMECs elicits an interferon response

To assess the susceptibility and permissiveness of primary LMECs to IAV infection, we inoculated cells with pandemic H1N1 2009 virus (pH1N1) or seasonal H1N1 or H3N2 viruses isolated in 2019. To confirm that infection of endothelial cells is equally efficient when cells are directly exposed to virus, or when there is a membrane in between the virus inoculum and LMECs, we cultured LMECs either at the apical or basolateral side of a filter in a transwell system. In LMECs cultured at the apical or basolateral side, no increase of infectious virus titers was detected in the apical or basolateral compartments (Fig 1A and B). Virus nucleoprotein (NP) antigen was detected in some virus-inoculated LMECs 24 hours post-inoculation (hpi) by immunofluorescence (IF) staining in both experimental set-

ups (Fig 1C and D). Flow cytometry analysis revealed 1–2% IAV-infected cells at 24 hpi and <1% at 72 hpi of NP⁺ LMECs. pH1N1 virus infected significantly more LMECs than H1N1 and H3N2 viruses at 24 (~2% versus ~1%) and 72 hpi (~1% versus 0.3%), respectively, (Fig 1E). To confirm infection of LMECs, we lysed infected cells and quantified viral RNA genome copies by quantitative real-time PCR (qRT-PCR) over time and showed that intracellular viral genome replication peaked at 24 hpi for all three viruses, after which the amount of genome copies declined (Fig 1F).

To evaluate whether infection of LMECs induced an antiviral immune response, we inoculated LMECs with pH1N1, H1N1, or H3N2 virus and measured RNA expression of *IFN-β*, *IFN-λ* and interferon-stimulated gene (ISG) *IFIT1* 24 hpi (Fig S1). As a positive control, we exposed LMECs either to recombinant IFN-β, IFN-λ, or to Toll-like receptor 3 agonist polyinosinic:polycytidylic acid (poly I:C), mimicking double-stranded RNA, either in the supernatant or via transfection. Upon pH1N1, H1N1, or H3N2 virus infection, *IFN-β*, *IFN-λ*,

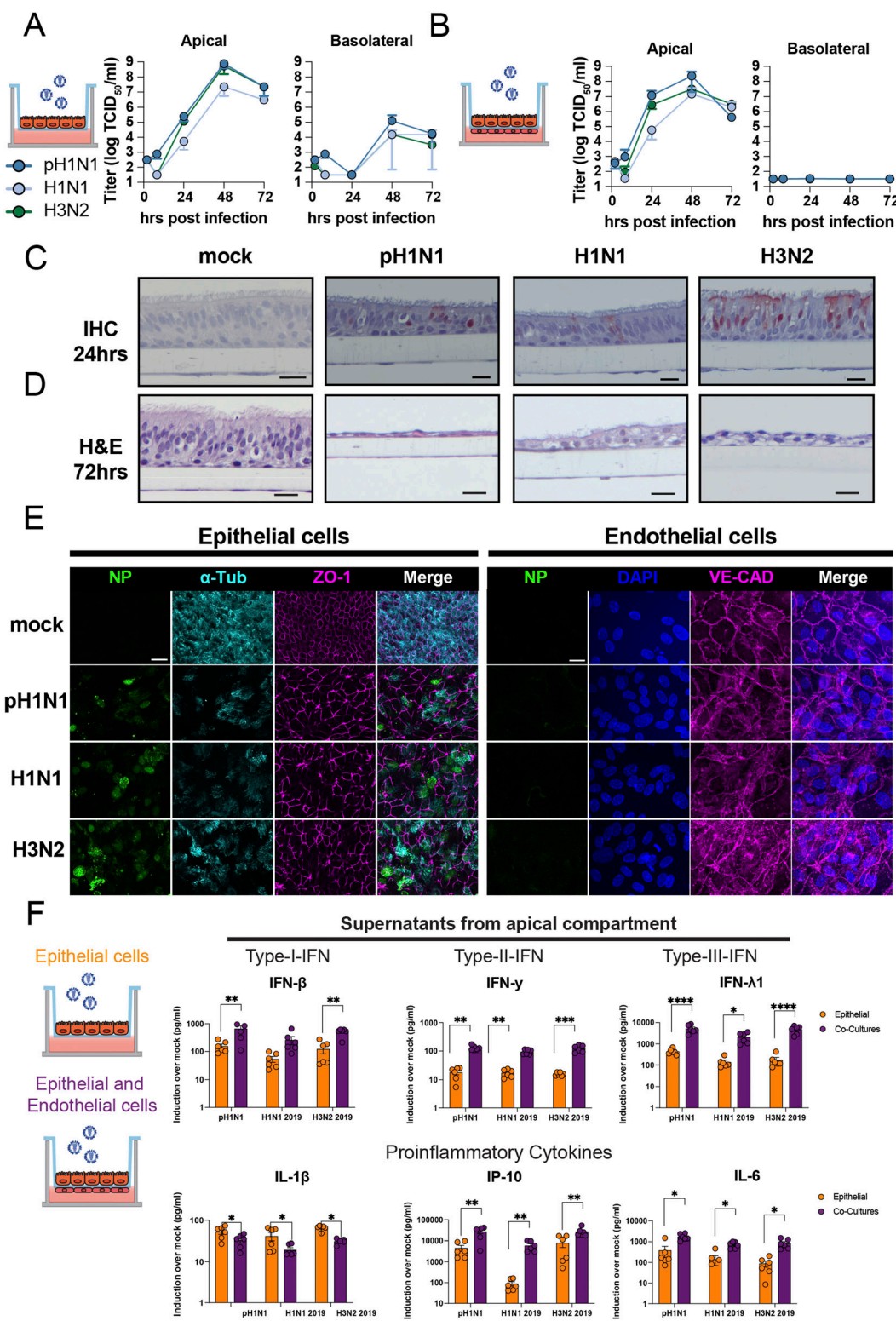

**Figure 2. Influenza A virus infection and cytokine profiling of the apical compartment of epithelial cell mono-cultures or in co-culture with endothelial cells.**
**(A, B)** Well-differentiated airway organoids at air–liquid interface (AO at ALI) in mono-culture or (B) in co-culture with lung microvascular endothelial cells (LMECs) were inoculated with pH1N1, H1N1, or H3N2 virus at MOI 1. At the indicated timepoints, virus titers were determined in the supernatants of the apical and basolateral compartments. **(C)** Detection of influenza A virus (IAV) nucleoprotein (NP) by immunohistochemistry (IHC) of the AO at ALI-LMEC co-cultures 24 h post-inoculation. **(D)** Hematoxylin and eosin (H&E) staining of the co-cultures 72 h post-inoculation (scale bar 20 $\mu$m). **(E)** At 72 h post-inoculation, well-differentiated AO at ALI were stained for IAV NP (green), the cilia marker acetylated-$\alpha$-tubulin ($\alpha$-Tub, cyan) and the tight junction marker zona occludin-1 (ZO-1, magenta) on the apical compartment

and *IFIT1* were significantly up-regulated compared to the mock control. We also investigated if intracellular genome replication induces a type I and type III interferon response. Under the positive control conditions, *IFN-β*, *IFN-λ*, and *IFIT1* were significantly induced, except after IFN-λ treatment where only *IFIT1* and *IFN-λ* were significantly induced. This suggests that the expression of LMECs interferon α/β receptor (IFNAR) is higher than IFNLR in LMECs. In conclusion, we detected intracellular viral genome replication of pH1N1, H1N1, and H3N2 viruses in LMECs; however, we did not detect an increase of infectious virus progeny in the supernatant, which is a sign of abortive infection (Fig 1A and B).

### Replication of influenza A viruses in epithelial mono-cultures and epithelial–endothelial co-cultures induces a robust cytokine response

Epithelial cells are the first cells to be infected by IAV in the lung. We used AO at ALI as our source of respiratory epithelial cells and validated that this was a pseudostratified layer containing ciliated and goblet cells (Fig S2A and B, respectively). Next, we developed a co-culture model using transwells containing AO at ALI at the apical side and LMECs at the basolateral side of the transwell. pH1N1, H1N1, or H3N2 virus inoculation of epithelial cells resulted in efficient virus replication in the mono-epithelial cultures and in the co-cultures when measured in the apical compartment (Fig 2A and B). The virus replication in epithelial mono-cultures compared with co-cultures showed that replication was slightly faster in co-cultures (Fig S3). In the basolateral compartment, infectious virus was only detected at 48 hpi in the mono-cultures, which was associated with a reduction in the transepithelial electrical barrier (TEER) (Figs 2A and S2C). No infectious virus particles were detected in the basolateral compartment of the co-cultures (Fig 2B). IAV infection of ciliated epithelial cells could be detected at 24 hpi with minimal visible cell damage using a hematoxylin and eosin staining (Fig 2C and D). At 72 hpi, flattening of the epithelial cell layer was observed associated with hypertrophic epithelial cells and loss of ciliated epithelial cells. Loss of cilia was confirmed using an IF staining for acetylated α-tubulin, and cultures were depleted of cilia at 72 hpi (Fig 2E). In addition, we observed loss of the tight-junction marker Zona Occludin-1. In IAV-infected LMECs, we detected changes in the expression of vascular endothelial-cadherin, which was more diffuse than uninfected cells where it was predominantly observed on the cell surface. In the LMECs, there were also some structural changes visualized by vascular endothelium-cadherin (Fig 2E).

Next, we evaluated the pro-inflammatory cytokine response of infected epithelial cells in mono-culture or in co-culture in the apical compartment on protein level. We mainly detected pro-inflammatory cytokines (IL-6, IL-1b, and IP-10) and IFNs (IFN-β, IFN-γ, and IFN-λ1), in both mono-cultures and co-cultures. A more pronounced cytokine response was detected in the co-cultures compared to the mono-cultures at 24 hpi (Fig 2F). At 72 h, no differences were observed between the cultures (Table S1). In conclusion, the AO at ALI were highly susceptible and permissive to seasonal and pandemic IAV, and replication was associated with a robust pro-inflammatory cytokine response and breakdown of epithelial barrier integrity.

### Endothelial cells are not productively infected by influenza A viruses in epithelial–endothelial co-cultures but show an enhanced inflammatory response compared with mono-cultures

In subsequent experiments, we focused on the LMECs in our epithelial–endothelial co-culture model. Even though all IAV robustly replicated in epithelial cells (Figs 2 and S3), we did not detect IAV NP in LMECs in these co-cultures at 24 or 72 h post epithelial cell inoculation (Fig 2C and E). This was in contrast to the LMECs in mono-culture where we found evidence for virus infection with immunofluorescence (Fig 1C and D). Next, we compared the infected LMECs in co-culture to LMEC mono-cultures and found significantly fewer infected cells in the co-cultures compared with infected LMEC mono-cultures (Fig 3A). In addition, significantly less viral RNA was detected in the LMECs of the co-cultures compared with inoculated LMECs in mono-cultures (Fig 3B). When LMECs were directly exposed to pH1N1, H1N1, or H3N2 virus by basolateral inoculation 24 h after apical inoculation of epithelial cells, we rarely detected IAV NP in LMECs (Fig S4).

Next, we wanted to investigate the pro-inflammatory response of AO at ALI and LMECs in co-culture. No large differences in the up-regulation of *IFIT1* or *IFN-β* gene expression were observed following pH1N1 virus infection when comparing lysates of apically infected AO at ALI or LMECs. However, *IFN-λ* expression was significantly higher in the endothelial cells compared to the epithelial cells (Fig S5). Finally, we investigated the cytokine protein concentrations secreted by endothelial cells in co-culture or in mono-cultures in the basolateral compartment (Table S1). Protein concentrations of interferons (IFN-β, IFN-λ, and IFN-γ) and pro-inflammatory cytokines (IL-1β, IP-10, and IL-6) were higher upon virus infection in all cultures (Fig 4). Epithelial cells alone secreted mainly IFN-β and IFN-λ in the basolateral compartment but no IFN-γ, whereas endothelial cells alone were also capable of producing IFN-γ. Both epithelial cells and endothelial mono-cultures produced IP-10 and IL-6 in the basolateral compartment, although endothelial cells seemed to contribute more to this production. Notably, we detected higher concentrations of cytokines in the co-cultures compared to mono-cultures, which was more than the sum of the concentrations of cytokines produced in the mono-cultures. This effect was not only most pronounced for IL-1β but also observed for IFN-β, IFN-γ, and IFN-λ. For other cytokines, we measured either above or below (IL-10) the limit of detection or showed no significant induction

of the transwell. The basolateral compartment containing the LMECs was stained for IAV NP (green) and the endothelial cell marker vascular endothelial-cadherin (VE-CAD, magenta). In both cases, the nuclei were visualized with Hoechst (blue, scale bar: 20 μm). **(F)** Epithelial cells (AO at ALI) or endothelial–epithelial co-cultures were inoculated with pH1N1, H1N1, or H3N2 virus at MOI 1. At 24 h post-inoculation, cytokines were measured in the apical compartment, and protein concentrations were determined using the LEGENDplex assay. Data represented here show individual data points of cytokines derived from three independent experiments performed in biological duplicates, and the mean ± SD is depicted. Mock protein concentration of each condition was subtracted from the protein concentrations measured in the virus inoculated cultures. Statistical significance was determined with *t* test (*<0.05, **<0.01, ***<0.005, ****<0.001).

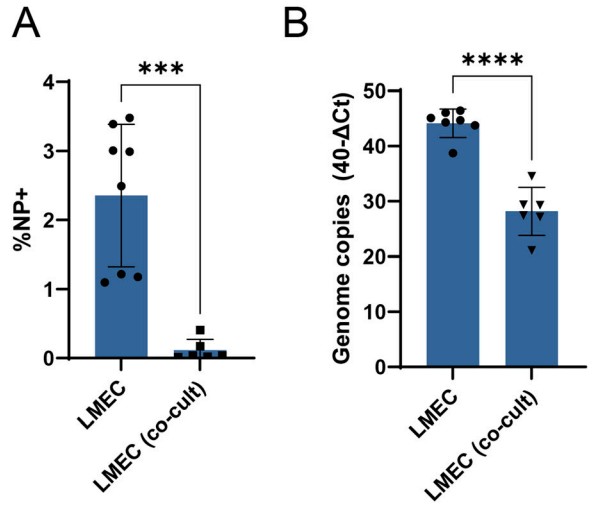

**Figure 3. Quantification of influenza A virus infection in endothelial mono-cultures or in epithelial–endothelial co-cultures.**
**(A)** Percentage of infection and **(B)** intracellular viral genome copies in LMEC mono-cultures compared with co-cultures were determined by flow cytometry or quantitative real-time PCR at 24 h post-inoculation. Data represented here show pooled data of either infection percentage or virus titers derived from three independent experiments performed in biological duplicates, and the mean ± SD is depicted. A t test was used to compare groups (*<0.05, **<0.01, ***<0.005, ****<0.001).

upon IAV infection (TNF-α, IL-12p70, IFN-λ2/3, GM-CSF, and IFN-α2) (Table S1). There were no clear differences among the different viruses with respect to their cytokine profile.

### Endothelial cells in in vivo pH1N1–infected ferret lungs and ex vivo pH1N1–infected human lungs express IL-6 mRNA

One of the main pro-inflammatory cytokines produced by LMECs in the co-culture model was IL-6. To confirm our in vitro findings in an in vivo model, we investigated if endothelial cells express IL-6 mRNA in experimentally inoculated ferrets. Therefore, we measured IL-6 mRNA by in situ hybridization (ISH) in the lungs of three ferrets inoculated with pH1N1 virus and in ex vivo human lung biopsies infected with pH1N1 (22, 23). In all investigated ferret lungs and human lungs, IL-6 mRNA transcripts were detected using ISH in von Willebrand factor (VWF) positive endothelial cells detected by immunohistochemistry in regions associated with lesions (IHC, Figs 5A and B and S6). From this analysis, we cannot formerly conclude that the IAV infection was directly associated with the expression of IL-6 mRNA in endothelial cells as no uninfected tissues were included. However, we feel that these data support the relevance of the newly established in vitro co-culture model and provides evidence that these data possibly are translatable to our in vitro finding that endothelial cells contribute to IL-6 secretion also to the in vivo setting.

## Discussion

In this study, we assessed the role of pulmonary endothelial cells in the pro-inflammatory responses upon IAV infection in a co-culture model consisting of AO at ALI and LMECs. We measured higher levels of pro-inflammatory cytokines in the basolateral compartment when endothelial cells were co-cultured with IAV-inoculated epithelial cells compared with infected endothelial or epithelial mono-cultures. This response was not associated with productive infection of the endothelial cells. One of the cytokines predominantly secreted by LMECs in the basolateral compartment was IL-6, which was also detected in endothelial cells in vivo in pH1N1 virus–inoculated ferrets and in ex vivo inoculated–human lung biopsies.

IAV infection of well-differentiated primary epithelial cells, either directly isolated from airway epithelial cells or stem cell based, is well characterized (24, 25). Similar to our results, IAV was described to replicate efficiently in epithelial cells, cause damage to the epithelial barrier, and induce robust amounts of pro-inflammatory cytokines (such as IL-1b, IL-6, IP-10, type I/III IFNs) in the apical compartment (24, 26, 27, 28, 29). Interestingly, here, we show that when epithelial cells were co-cultured with endothelial cells, the replication kinetics of all three IAV strains appeared to be slightly faster, and the levels of pro-inflammatory cytokines were increased compared with epithelial mono-cultures. This suggests that the interaction between epithelial and endothelial cells in the respiratory tract influences the replication of and response to IAV infection.

The role of endothelial cells in the pathogenesis of influenza in humans is not well understood. Endothelial cells are rarely infected in vivo by zoonotic, pandemic, and seasonal IAV, as measured by detection of viral antigen (10, 22, 30, 31). Our study shows that small percentages of endothelial cells can be infected abortively by seasonal and pandemic influenza viruses when directly exposed to the viruses. Furthermore, interaction with infected epithelial cells might induce (at least in part) an antiviral state in endothelial cells. In line with our data, other in vitro studies on endothelial cells show low infection efficiency and abortive replication with seasonal or pandemic IAV (11, 14, 16, 21, 22, 31). In contrast to the low percentage of infected endothelial cells measured by virus antigen, single-cell RNA seq of PR8 (H1N1)-infected mice lungs showed a high prevalence (~50%) of viral RNA copies in pulmonary endothelial cells which correlated with high transcripts of ISGs (18). We also detected an induction of IFNs, ISGs, and other cytokines in LMECs harvested from mono-cultures or co-cultures. This induction was most pronounced in co-cultures, in absence of detectable IAV antigen but presence of viral RNA. These data strengthen the hypothesis that abortive viral replication in LMECs induces an antiviral and inflammatory response likely through the detection of viral RNA by PRRs.

Several mechanisms can explain the induction of antiviral and inflammatory responses in endothelial cells when cultured together with infected epithelial cells. First, an abortive infection in LMECs, measured by qRT-PCR, triggered an IFN response likely due to activation of PRRs. Our data show that IAV breaks down the TEER of AO at ALI and that endothelial cells can subsequently become abortively infected. Second, the transwells have pores (0.4 μm) which allow exposure of endothelial cells to cytokines produced by epithelial cells (19). Taken together, our results suggest that endothelial cells contribute to the inflammatory response to IAV infection by sensing viral RNA and/or epithelial–endothelial

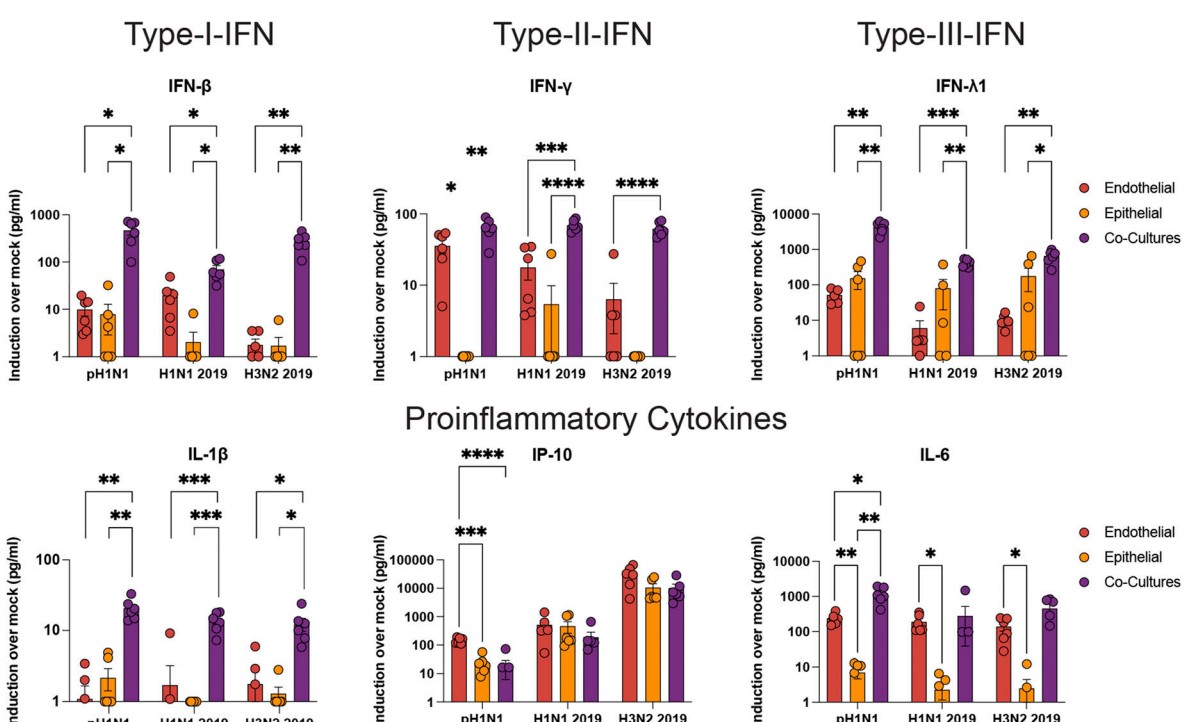

**Figure 4. Detection of cytokines produced by endothelial cells in the basolateral compartments of mono- or co-cultures.**
Lung microvascular endothelial cells, epithelial cells (airway organoids at air–liquid interface) or endothelial–epithelial co-cultures were inoculated with pH1N1, H1N1, or H3N2 virus at MOI 1. At 24 h post-inoculation, cytokine protein concentrations were measured in the basolateral compartment using a LEGENDplex assay. Data represented here show individual data points of cytokines derived from three independent experiments performed in biological duplicates, and the mean ± SD is depicted. Mock protein concentration of each condition was subtracted from the protein concentrations measured in the virus-inoculated cultures. Statistical significance was determined with one-way ANOVA and each group was compared with each other (*<0.05, **<0.01, ***<0.005, ****<0.001).

crosstalk and that both cell types are required when studying the context of IAV-induced inflammation in the lungs (10, 11, 19).

Endothelial cells line the blood vessels, and cytokines produced by these cells will directly enter the bloodstream, potentially contributing to systemic inflammation in the blood. High levels of IL-6, IFN-γ and IP-10 in the blood and bronchoalveolar lavage—cytokines that were produced to high levels by LMECs when co-cultured with epithelial cells—have been described as a biomarker for severe IAV disease in humans and mice (32, 33, 34, 35, 36). In addition, we showed that IL-6 mRNA is expressed by endothelial cells in vivo in ferrets and in ex vivo human lungs infected with pH1N1. As endothelial cells are prominent cells in the lung (30%) (18, 37), it is likely that endothelial cells are important contributors to the systemic inflammatory responses observed during severe influenza.

In conclusion, the described co-culture model suggests that abortively IAV-infected endothelial cells can fuel pro-inflammatory responses. Intervening or reducing the pro-inflammatory responses in endothelial cells could potentially reduce morbidity and mortality associated with influenza.

## Materials and Methods

### Primary cells and cell lines

Madin–Darby canine kidney (MDCK) cells were maintained in Eagle's Minimal Essential Medium (EMEM; Lonza) supplemented with 10% FCS, 100 IU/ml penicillin, 100 µg/ml streptomycin, 2 mM glutamine,

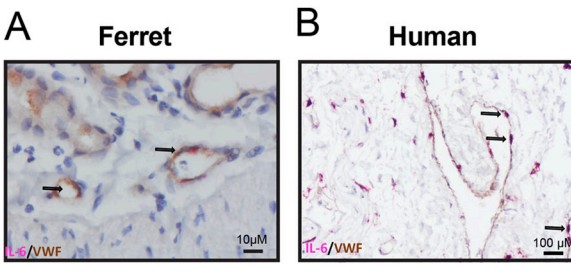

**Figure 5. Endothelial cells in pH1N1-inoculated ferret and human lungs express IL-6 mRNA.**
**(A)** IL-6 production by endothelial cells was assessed in lung sections of a ferret inoculated with pH1N1 virus (1-d post-inoculation) and **(B)** lung sections of human lung biopsies inoculated with pH1N1 virus (1-d post-inoculation) by in situ hybridization for IL-6, followed by immunohistochemistry using an antibody for endothelial cells (von Willebrand factor, VWF). Arrows indicate cells that are positive for IL-6 and VWF.

1.5 mg/ml sodium bicarbonate (1 mM), 10 mM HEPES, and 0.1 mM nonessential amino acids. The medium was refreshed every 3 to 4 d, and cells were passaged at >90% confluence. Primary human LMECs were purchased at passage 3 from PromoCell-PromoKine (#C-12285) and cultured in 1% gelatin-coated cell cultures flasks with endothelial cell growth medium MV-2 kit according to the manufacturer's protocol (#C-22121; PromoCell-PromoKine). LMECs were only used up to passage 10 to ensure organ specificity. The cells were routinely checked for the presence of mycoplasma. Polyinosinic:polycytidylic acid (poly I:C) was purchased from InvivoGen and dissolved in water to a stock concentration of 1 mg/ml.

### Viruses

Pandemic H1N1 virus (pH1N1 2009, A/Netherlands/602/2009) was isolated from a patient who visited Mexico. The isolate was propagated three times in MDCK cells. Seasonal H1N1 and H3N2 viruses were isolated in 2019 and were kindly provided by the National Influenza Centrum (Table 1). The received isolates were propagated once in MDCK cells.

### Human primary airway culture and differentiation

Human airway organoid cultures (38) were generated and differentiated based on published protocols (38, 39). To obtain differentiated organoid-derived cultures at air–liquid interface, organoids were disrupted into single-cell suspension with TrypLE Express and seeded on Transwell membranes (Corning) coated with rat tail collagen type I (Thermo Fisher Scientific) in airway organoid medium and complete base medium at a 1:1 ratio. When a monolayer was confluent (2–3 d), the apical medium was removed and the basolateral medium was replaced with complete base medium. The cultures were differentiated for 4–8 wk with fresh medium every 5 d. Differentiation was visually confirmed by the production of mucus and ciliary movements or by antibody staining for tight junctions (zona occludens-1) and cilia (acetylated $\alpha$-tubulin) and assessment by confocal laser scanning microscopy (LSM 700; Zeiss).

**Table 1. Subtype and clade of 2019 viruses.**

| Virus | Subtype | Clade | Dutch identification number |
|---|---|---|---|
| H3N2 2019 | AH3N2 | 3C.3a | 19A01618 |
| H1N1 2019 | AH1N1pdm09 | 6B.1A5A | 19A01659 |
| H1N1 2009 | AH1N1pmd09 | 6B.1 | GISAID Isolate ID: EPI_ISL_31217 |

### Epithelial and endothelial co-cultures and viral infections

The co-culture model of primary human epithelial and endothelial cells was established as follows. After full differentiation of human primary airway cultures on air–liquid interface, the transwell inserts were inverted and coated with 150 µl 1% gelatin at 37°C for 1 h. After the coating, the basolateral side of the transwell was washed once with PBS and 150,000 LMECs/well were seeded in 100 µl MV2 medium in the basolateral compartment. To allow attachment of LMECs transwells were incubated at 37°C for 1 h. Afterward, the basolateral compartment was filled with 750 µl PneumaCult-ALI Medium (#05001; STEMCELL Technologies), mixed with 750 µl MV2 medium. For virus infection of the primary airway cultures at air–liquid interface, first, the apical cells were incubated with PBS containing $Mg^{2+}$ and $Ca^{2+}$ for 20 min, and then the mucus was washed away by pipetting up and down. This process was repeated two times. According to the MOI, virus stocks were diluted in PBS containing $Mg^{2+}$ and $Ca^{2+}$ and 150 µl of PBS, and virus were added to the apical side. In case of epithelial mono-culture experiments, the medium of basolateral compartment consisted of 1:1 mixture MV2 medium and PneumaCult, similar to the co-cultures. After 1 h of attachment, the cells were washed three times with PBS containing $Mg^{2+}$ and $Ca^{2+}$. For infection of the endothelial cells, the transwell inserts were inverted, and the LMECs were exposed to the corresponding MOI of virus diluted in MV2 medium. After one hour of incubation at 37°C, the basolateral side was washed three times with PBS, and 750 µl PneumaCult-ALI Medium mixed with 750 µl MV2 medium was added freshly.

### Transepithelial resistance measurements

A monolayer of epithelial cells grown on Transwell filters (0.4-$\mu$m-diameter pores; Corning) were infected with pH1N1, H1N1, and H3N2 viruses with MOI 1 or left uninfected. The monolayer was analyzed for transepithelial resistance with an EVOM voltohmmeter (World Precision Instruments) with an STX-2 chopstick electrode at the indicated timepoint. The measured values were calculated by multiplying the electrical resistance by the area of the filter.

### Quantitative PCRs

Total RNA was isolated from cells using the High Pure RNA Isolation Kit (Roche) on a MagNA Pure machine and used for qRT-PCR. In short, 5 µl total RNA was amplified in a mix containing 5 µl of TaqMan Fast Virus 1-Step Master Mix (Life Technologies), 1 µl primer–probe mix for $\beta$-actin, IFN-$\beta$1, IFN-$\lambda$1, IFIT1 (Thermo Fisher Scientific), and 9 µl distilled water. The qRT-PCR temperature profile was 5 min at 50°C, 20 s at 95°C, followed by 40 cycles of 3 s at 95°C and 30 s at 60°C. For determining the viral genome copies, 7 µl total

**Table 2. Antibodies used for immunofluorescence microscopy.**

| Antibodies | Cat # | Final concentration | Dilution | Manufacturer |
|---|---|---|---|---|
| NP | EBS-I-047 | 2.0 μg/ml | 1:1,000 | EVL Laboratories |
| VE-CAD | AF938 | 2.0 μg/ml | 1:100 | R&D Systems |
| Acetylated-α-tubulin | SC-23950 | 2.0 μg/ml | 1:100 | Santa Cruz Biotechnology |
| ZO-1 | 339100 | 5.0 μg/ml | 1:100 | Life Technologies/Invitrogen |
| Hoechst | 62249 | 2 μM | 1:10,000 | Life Technologies/Invitrogen |

RNA was amplified in a mix containing 5 μl of TaqMan Fast Virus 1-Step Master Mix (Life Technologies), 0.4 μl primer–probe mix for IAV (40), and 9.6 μl distilled water. The qRT-PCR temperature profile was 5 min at 50°C, 20 s at 95°C, followed by 45 cycles of 3 s at 95°C and 30 s at 60°C. The fold induction or genome copies fold change to input was calculated with the double delta Ct method (41).

## Titration

Virus titers were determined by endpoint dilution on 30,000 MDCK cells/well-plated in 96-wells. 10-fold serial dilutions of cell supernatant in technical triplicate were prepared in infections medium consisting of EMEM supplemented with 100 IU/ml penicillin, 100 μg/ml streptomycin, 2 mM glutamine, 1.5 mg/ml sodium bicarbonate, 10 mM HEPES, 1× (0.1 mM) nonessential amino acids, and 1 μg/μl tosylsulfonyl phenylalanyl chloromethyl ketone–treated trypsin (Sigma-Aldrich). Before adding the titrated supernatants, the MDCK cells were washed once with plain EMEM medium to remove residual FCS. 100 μl of the titrated supernatants were used to inoculate MDCK cells and after 1-h, infectious supernatant was removed and 200 μl fresh infection medium was supplemented. 3 d after infection, the supernatants of the MDCK cells were tested for agglutination. For that, 25 μl of supernatant was mixed with 75 μl 0.33% turkey red blood cells and incubated for 1 h at 4°C. Infection titers were calculated according to the method of Kärber and expressed as TCID50/ml (42).

## Immunofluorescence labeling

Cells on transwells were fixed by adding 10% formalin (1 ml to the basolateral side and 500 μl apical side) to the transwells for 30 min. Afterward cells were washed three times with PBS and permeabilized with PBS containing 1% Triton X-100 followed by a 30 min incubation in blocking solution consisting of PBS supplemented with 0.5% Triton X-100 and 1% BSA at RT. With a scalpel, the transwells were cut out of the plastic frame, and the membrane was cut in half and washed three times in PBS. Primary antibody concentrations that were used according to the Table 2, were diluted in blocking solution and incubated at RT for 1 h. Following a washing step where membranes were dipped three times in PBS, they were stained with the corresponding secondary antibodies, and Hoechst to visualize the nuclei, at RT for 1 h. Stained membranes were afterward washed three times in PBS and once in water and mounted in ProLong Antifade Mountant. Membranes were imaged using a Zeiss LSM 700 confocal microscope.

## Multiplexed bead assay for cytokine profiling

Cytokine concentration of the apical and basolateral compartments was measured using the human antivirus response panel (13-plex) kit (LEGENDplex; BioLegend). The kit was used according to the manufacturer's manual. The data were analyzed with flow cytometry and final analysis of the concentration was performed using LEGENDplex analysis software v8.0. For graphs in Figs 2 and 4, the representative data show individual data points of cytokines derived from three independent experiments performed in biological duplicates. We subtracted the mock protein concentration from the virus infected condition to show the induction of cytokines upon virus infection over mock-treated infection. The raw values of the protein concentrations of each biological duplicate under all three independent experimental conditions can be found in Table S1.

## FACS staining

LMECs and MDCK were first washed with PBS and then released with trypsin and collected in a V-bottom plate. The cells were permeabilized using cytofix/cytoperm (BD Biosciences) for 20 min and blocked with 10% normal goat serum for 30 min. The cells were incubated at 4°C for 30 min with an unconjugated antibody for IAV NP (HB65, 8.0 μg/ml) followed by a PE-conjugated goat anti-mouse secondary antibody (DAKO) and measured on a flow cytometer (BD FACSLyric).

## Pathologic examination

Membranes with co-cultured epithelial and endothelial cells were collected and fixed in 10% neutral-buffered formalin for 30 min, cut in half, and embedded in paraffin. For examination by light microscopy, 3 μm sections were deparaffinized and stained with hematoxylin and eosin (HE). Sections were also stained with periodic acid–Schiff for detection of mucoid cells according to standard methods.

## Influenza IHC

For detection of IAV nuclear protein (NP) in co-cultured epithelial and endothelial cells, 3 μm formalin-fixed, paraffin-embedded cross sections of the co-culture membranes were deparaffinized and rehydrated. NP antigen was retrieved by incubating sections briefly for 2 min in 0.1% protease at 37°C. The slides were then washed with PBS/0.05% Tween 20 and incubated with mouse IgG2a-

anti–IAV NP (Clone Hb65; EVL Laboratories), 5 µg/ml or mouse IgG2a isotype control (MAB003; R&D Systems), 2.5 µg/ml in PBS/0.1% BSA for 1 h at RT. After washing, sections were incubated with horse-radish peroxidase–labeled goat anti-mouse IgG2a (STAR133P; Serotec) 10 µg/ml in PBS/0.1% BSA for 1 h at RT. Peroxidase activity was revealed by incubating slides in 3-amino-9-ethylcarbazole (AEC) (Sigma-Aldrich) for 10 min, resulting in a bright red precipi-tate, followed by counterstaining with hematoxylin. A lung section from an experimentally influenza (pH1N1)–inoculated ferret was used as a positive control.

### In Situ Hybridization (ISH) for the analysis of IL-6 expression in pH1N1-infected ferrets and pH1N1-infected ex vivo human lungs

For determination of expression of IL-6 by endothelial cells upon influenza virus infection, lung sections of three ferrets inoculated with pH1N1 virus (1 day post-inoculation [dpi] (22)) and human lung slices (approval license MEC-2008-307) inoculated with pH1N1 (1 dpi) (23) were double stained by ISH for IL-6 using the RNAScope platform, followed by IHC using an antibody for endothelial cells (von Willebrand factor, VWF). RNA probes were designed by ACD Biotechne for ferret IL-6 (300031) and for human IL-6 (310371). First, ISH was performed on 3 µm formalin-fixed, paraffin-embedded lung sections using RNAScope Reagent Kit v2–RED (322350) as described by the manufacturer, up to amplification step 6. Slides were then washed in PBS/0.05% Tween 20 and incubated with rabbit anti-human VWF (A008229-2; DAKO/Agilent tech) 1/500 or rabbit IgG isotype control (AB-105-C; R&D Systems) 1/100 in PBS/ 0.1% BSA for 1 h at RT. After washing, sections were incubated with biotinylated goat polyvalent anti-mouse IgG (H + L) and anti-rabbit IgG (H + L) (TP-125-HL; LabVision) for 10 min at RT, followed by horseradish peroxidase–labeled streptavidin (D0397; DAKO/ Agilent tech) 1/300 in PBS/0.1% BSA for 30 min at RT. Slides were washed, and IL-6 RNA molecules were visualized as red chromogenic dots according to the RNAScope protocol. Peroxi-dase activity from VWF IHC was revealed by incubating slides in 3,3'–diaminobenzidine-tetrachlorohydrate (DAB) (Sigma-Aldrich) for 3–5 min, resulting in a brown precipitate. Slides were coun-terstained with Gill's hematoxylin, dried at 60°C for 5–10 min and mounted with Ecomount.

### PolyI:C treatment

Subconfluent LMECs were plated on 24-wells coated with 1% gel-atine. On the next day, LMECs were transfected with 10 or 1 µg/ml polyI:C with Lipofectamine 3000 (Invitrogen) according to the manufacturer's protocol. After 24 h, cells were harvested in lysis buffer, and RNA was extracted with the MagNA Pure machine.

### Statistical analysis

Statistical differences between experimental groups were de-termined as described in the figure legends. *P*-values of ≤0.05 were considered significant. Graphs and statistical tests were made with GraphPad Prism version 9. Figures were prepared with Adobe Illustrator CC2019, Adobe Photoshop CC2019, and BioRender.

## Supplementary Information

## Acknowledgements

D van Riel is supported by fellowships from the Netherlands Organization for Scientific Research (VIDI contract 91718308) and a EUR fellowship.

### Author Contributions

L Bauer: conceptualization, resources, data curation, formal anal-ysis, supervision, validation, investigation, visualization, method-ology, and writing—original draft, review, and editing.
LC Rijsbergen: conceptualization, resources, data curation, formal analysis, validation, investigation, visualization, and writing—original draft, review, and editing.
L Leijten: resources, data curation, and investigation.
FFW Benavides: data curation and formal analysis.
D Noack, MM Lamers, BL Haagmans, RD de Vries, and RL de Swart: resources and writing—review and editing.
D van Riel: conceptualization, resources, formal analysis, supervi-sion, funding acquisition, investigation, and writing—original draft, review, and editing.

### Conflict of Interest Statement

The authors declare that they have no conflict of interest.

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
