## [Reviewer comments · Life Science Alliance]

Life Science Alliance

The pro-inflammatory response to influenza A virus infection is fueled by endothelial cells

Lisa Bauer, Laurine Rijsbergen, Lonneke Leijten, Feline Benavides, Danny Noack, Mart Lamers, Bart Haagmans, Rory de Vries, Rik De Swart, and Debby van Riel

DOI: <https://doi.org/10.26508/lsa.202201837>

Corresponding author(s): *Debby van Riel, Erasmus MC and Lisa Bauer, Erasmus MC*

Review Timeline:

Submission Date:	2022-11-18
Editorial Decision:	2022-12-19
Revision Received:	2023-03-16
Editorial Decision:	2023-03-30
Revision Received:	2023-04-05
Accepted:	2023-04-05

Scientific Editor: Novella Guidi

Transaction Report:

December 19, 2022

Re: Life Science Alliance manuscript #LSA-2022-01837

Dr. Debby van Riel
Erasmus Medical Center
Netherlands

Dear Dr. van Riel,

Thank you for submitting your manuscript entitled "The pro-inflammatory response to influenza A virus infection is fueled by endothelial cells" to Life Science Alliance. The manuscript was assessed by expert reviewers, whose comments are appended to this letter. We invite you to submit a revised manuscript addressing the Reviewer comments.

Thank you for this interesting contribution to Life Science Alliance. We are looking forward to receiving your revised manuscript.

Sincerely,

B. MANUSCRIPT ORGANIZATION AND FORMATTING:

Reviewer #1 (Comments to the Authors (Required)):

Authors established a co-culture system to investigate interplay in proinflammatory responses between human lung endothelial and epithelial cells following influenza A virus infection. Different cell types were propagated on opposing sides of Transwell inserts and viral infectivity, replication cytokine/chemokine and histopathological parameters were investigated against a small panel of human H1 and H3 viruses. Authors found that the presence of endothelial cells contribute to proinflammatory host responses despite not supporting robust replication of IAV themselves. Given the dynamic nature of host responses in multicellular tissue environments like the human respiratory tract, there is a need for novel and innovative cell culture approaches to better model these complexities outside of living hosts. While the study is somewhat descriptive, the manuscript rationale is clear, experiments are generally clearly presented, with results supported by the conclusions drawn. However, there are areas of the manuscript that would benefit from improved data presentation, contextualization, and focus.

Comments:

1. Lines 106-7, can the authors use immunostaining approaches to answer if this speculative statement (that LMECs express IFNAR but not IFNLR) is true, or are there studies in the literature that would further support this statement? As it stands in its current form it is rather speculative and probably not well-suited for the results.
2. Figure 1A and B, and Figure 2A, the error bars are very faint and challenging to visualize by the reader.
3. Lines 142-144, it would be very helpful to the reader to put the mono-culture graph in the main figure alongside Figure 2A to more clearly support the statement that mono-culture viral titers were higher relative to co-culture (it looks like there is space, and would provide the reader a more direct way to assess this point instead of jumping between main text and supplemental figures).
4. Figure 2E, please specify what actual mock pg levels were subtracted from values of virus-infected cells for each cytokine shown (in the methods or figure legend), so that the reader can better interpret the fold differences being presented.
5. Please better clarify in the titles for Figure 2 and 4 that cytokine levels were tested from the apical (Fig2) and basolateral (Fig4) compartments; right now this is only mentioned within the legend itself and since results are a bit different depending on the compartment sampled, having this key difference be more easily presented to the reader in the title would improve clarity.
6. Figure 4, suggest graphing all results uniformly on either linear or log scales; right now it appears two of the six panels use linear y axes and not log and as such the magnitude of differences presented is challenging to interpret when looking between different cytokines presented.
7. Lines 208-215: what do the authors think why virus infection was detectable in LMECs in mono-culture but not co-culture? Does this pertain to virus accessibility to the cellular surface or a different reason?
8. Figure 5 does not appear to add much to the paper; showing production of one cytokine at one time point from only one ferret and human slide each following infection with one virus is not very compelling and doesn't represent a strong way to end the paper. Suggest removing this figure outright (or including far more specificity about representative group sizes, sampling days, caveats about strain-specific influences, etc) and just adding references in the discussion to support other studies in the literature that have reported cytokine mRNA levels in ferrets and humans following IAV infection. If this finding is truly novel, it should be better contextualized and integrated throughout (beyond lines 338-332 as currently discussed in the discussion).
9. Line 291, "replication kinetics of all three IAV strains changed", please be more specific about the changes authors are alluding to here.
10. Please provide more specific information regarding the three different IAV studied here. Why do the authors think that the pdm09 virus infected LMECs to higher levels than the 2019 H1 or H3 viruses in this study (lines89-92)? What molecular changes are present between all three viruses that could assist interpretation of why the pdm09 virus? This data could be easily added to Table 1 with information presented from all three strains employed in the study.
11. Lines 404-405, what were the electrical resistance values of cultures at the time of viral infection, and did these values

change between mono-culture and co-culture conditions?

Reviewer #2 (Comments to the Authors (Required)):

This manuscript by Bauer and Rijsbergen uses an exciting model system combining differentiated airway epithelial cells and endothelial cells. As expected they find poor infection of seasonal of endothelial cells alone or in coculture. However they find reduced replication levels in epithelial cells cocultured compared to epithelial cells alone. Intriguingly they find differences in pro inflammatory cytokines and IFNs in coculture, better mimicking what happens in vivo. Overall then study is well designed and written and provides a strong platform to study epithelial cell-endothelial cell cross talk. The study could be improved by determining if IL-1b, IL-6, or IFNs are responsible for the differences in virus replication between single and co culture, although these experiments would likely be outside the scope of this manuscript. My only minor concern is that figure S3 should be moved into the main text (as a part of Fig 2 or 3) and would be improved by measuring total virus across the time course (area under the curve) and applying statistics to see if the viral load results are significant.

Reviewer #3 (Comments to the Authors (Required)):

The authors investigate the role of endothelial cells in the pathology of influenza virus and make an important contribution showing that while not all IAV can replicate in endothelial cells the abortive infection and/or the infection of nearby epithelial cells can turn the endothelial cells into producers of cytokine that could contribute to the disease pathology. The figures are very well made and the results are of high scientific quality.

We had a struggle in going through the result section as some results are coming back in different figures. Other results are shown but are not discussed in the discussion so it is unclear if these contribute to the paper. I believe the authors try to tell a chronological story, but perhaps better to separate the results on abortive IAV infection in one part and results on the cytokine secretion in a second part.

So my major comment is that the results part is badly structured and therefore also all the text is a struggle to get through. Maybe some results can be left out so there is no need to discuss them and a better focus is maintained.

1. In fig1 the authors show infection of endothelial cells that are at the basal or apical site of the membrane. The results are identical, which is to be expected. I don't understand why this experiment is in the paper. Later in the study the endothelial cells are always used at the basal side, so what is the value of the apical site grown endothelial cell infection experiment? As it is not discussed I believe this can be omitted from the paper.
2. the results in Fig 2E are already included in Fig 4, or what is the difference? Perhaps better to bring all cytokine profiling together
3. In fig 1 there is infection/replication of endothelial cell culture, but this also comes back in fig 3, can this not be combined?
4. Title of Fig2 is "Influenza A virus infection and cytokine profiling in epithelial cell cultures alone or in co-culture with endothelial cells." but I don't see the result on IAV replication in epithelial cells alone? The result on cytokine expression of infected epithelial cells is in Fig4, but are there also results on IAV replication in epithelial cells?
5. Perhaps it would be helpful to bring all results related to IAV replication in one part and the results related to cytokine expression in a second part.
6. In the conclusion the authors write "Interestingly, here we show that when epithelial cells were co-cultured with endothelial cells the replication kinetics of all three IAV strains changed," but we do not have info on replication in epithelial cells in the absence of endothelial cells. Or are the authors referring solely to replication in endothelial cells? There are several occasions where it is difficult to understand what is exactly meant. Please be concise.
7. when vRNA is determined, please always indicate if this is from cells or from cell culture medium, also in the figure legends.
8. in the intro the authors mention the "multibasic cleavage site". please explain and how it relates to this work
9. intro: "Suppression of early innate immune responses in endothelial cells and associated cytokine production greatly reduced the mortality of experimentally infected animals." I don't understand how immune suppression can reduce mortality, this seems a contradiction.
10. intro, something wrong with sentence "For example, high levels of pro-inflammatory responses detected in the blood, are associated with high morbidity in pandemic H1N1 viruses, either from 1918 or 2009. and zoonotic IAV infections in humans, or

in nonhuman primates"

11. when measuring cytokines it is not always indicated if this is RNA expression or protein that is measured. Please always indicate where needed.

line 107: after IFN-L treatment there is IFIT1 induction, so why does this suggest that IFNLR is not expressed? This is also not discussed are related to later in the paper. Is this then not relevant? Or I missed it perhaps.

line 108: "we do not detect infectious virus progeny in the supernatant" where is this shown?

line 110: "The intracellular genome replication 110 induces a type-I and type-III interferon response." - better to move this to line 104

general: I don't like "single cultures" perhaps better "mono cultures" as this better compares with co-cultures. I'm not sure if you always have to indicate that it is a mono culture, this is clear from context I believe.

abbreviation "TER", I believe the rest of the world uses TEER?

For figure 5 there are no uninfected controls

Response to the Reviewers #LSA-2022-01837

Reviewer #1 (Comments to the Authors (Required)):

Authors established a co-culture system to investigate interplay in proinflammatory responses between human lung endothelial and epithelial cells following influenza A virus infection. Different cell types were propagated on opposing sides of Transwell inserts and viral infectivity, replication cytokine/chemokine and histopathological parameters were investigated against a small panel of human H1 and H3 viruses. Authors found that the presence of endothelial cells contribute to proinflammatory host responses despite not supporting robust replication of IAV themselves. Given the dynamic nature of host responses in multicellular tissue environments like the human respiratory tract, there is a need for novel and innovative cell culture approaches to better model these complexities outside of living hosts. While the study is somewhat descriptive, the manuscript rationale is clear, experiments are generally clearly presented, with results supported by the conclusions drawn. However, there are areas of the manuscript that would benefit from improved data presentation, contextualization, and focus.

Comments:

1. Lines 106-7, can the authors use immunostaining approaches to answer if this speculative statement (that LMECs express IFNAR but not IFNLR) is true, or are there studies in the literature that would further support this statement? As it stands in its current form it is rather speculative and probably not well-suited for the results.

We agree with the reviewer that this statement is rather speculative with the data shown. We see a very mild induction of IFIT-1 (3-5 fold) and IFNL-1(7-10 fold) upon treatment of LMEC with recombinant human IFN-L. This argues in favor of lower expression of IFNLR compared to interferon α/β receptor on LMEC. *"This suggests that the expression of LMECs interferon α/β receptor (IFNAR) is higher compared to IFNLR in LMECs"* (Line 120-124)

2. Figure 1A and B, and Figure 2A, the error bars are very faint and challenging to visualize by the reader.

We enhanced the size of the error bars into Figure 1A and B, Figure 2A and B, and supplemental figure 3.

3. Lines 142-144, it would be very helpful to the reader to put the mono-culture graph in the main figure alongside Figure 2A to more clearly support the statement that mono-culture viral titers were higher relative to co-culture (it looks like there is space, and would provide the reader a more direct way to assess this point instead of jumping between main text and supplemental figures).

We have added the growth kinetics of the mono-cultures into the main figure as Panel 3A and added an appropriate description into the figure legend. *“(A) Well-differentiated airway organoids at air-liquid interface (AO at ALI) in mono-culture or (B) co-cultured with lung microvascular endothelial cells (LMECs) were inoculated with pH1N1, H1N1 or H3N2 virus at MOI 1. At the indicated timepoints virus titers were determined in the supernatants of the apical and basolateral compartments.”* (Line 649-650)

4. Figure 2E, please specify what actual mock pg levels were subtracted from values of virus-infected cells for each cytokine shown (in the methods or figure legend), so that the reader can better interpret the fold differences being presented.

We clarified the calculations better in the material and methods. As mentioned we subtracted the mock protein concentration from the protein concentrations measured after virus infection. To clarify that more, we added an additional comment in the Material and Methods part: *“For graphs in Figure 2 and Figure 4, the representative data show individual data points of cytokines derived from three independent experiments performed in biological duplicates. We subtracted the mock protein concentration from the virus infected condition to show the induction of cytokines upon virus infection over mock treated infection. The raw values of the protein concentrations of each biological duplicate in all three independent experimental conditions can be found in supplement table 1.”*(Line 401-403)

5. Please better clarify in the titles for Figure 2 and 4 that cytokine levels were tested from the apical (Fig2) and basolateral (Fig4) compartments; right now this is only mentioned within the legend itself and since results are a bit different depending on the compartment sampled, having this key difference be more easily presented to the reader in the title would improve clarity.

We specified in more detail in which compartment the protein concentrations were measured and we added an additional clarification into Figure 2 and Figure 4.

6. Figure 4, suggests graphing all results uniformly on either linear or log scales; right now it appears two of the six panels use linear y axes and not log and as such the magnitude of differences presented is challenging to interpret when looking between different cytokines presented.

As recommended by the reviewer we now displayed all results on log scales in Figure 4.

7. Lines 208-215: what do the authors think why virus infection was detectable in LMECs in mono-culture but not co-culture? Does this pertain to virus accessibility to the cellular surface or a different reason?

Our data show that endothelial cells are protected from infection, and not that endothelial cells are not exposed to virus. First, we show that virus infection of the epithelial mono-cultures results in disruption of TEER and virus diffuses through the trans-well as shown in Figure 2A. Furthermore, in the experiments from Figure S4 we exposed endothelial

cells co-cultured with infected epithelial cells directly to virus, which did not result in the detection of NP positive endothelial cells. It is likely that the cytokines released from infected epithelial cells results in an antiviral state of LMECS. This resembles the *in vivo* situation where epithelial cells are the primary target for influenza A viruses.

8. Figure 5 does not appear to add much to the paper; showing production of one cytokine at one time point from only one ferret and human slide each following infection with one virus is not very compelling and doesn't represent a strong way to end the paper. Suggest removing this figure outright (or including far more specificity about representative group sizes, sampling days, caveats about strain-specific influences, etc) and just adding references in the discussion to support other studies in the literature that have reported cytokine mRNA levels in ferrets and humans following IAV infection. If this finding is truly novel, it should be better contextualized and integrated throughout (beyond lines 338-332 as currently discussed in the discussion).

We apologise for not clearly stating that we analysed 3 individual pH1N1 virus inoculated ferrets, and that we observed IL-6 in endothelial cells of all three ferrets. This has been adjusted in the materials and methods (Line 438) and in the results (Line 203-205). Furthermore, we have included pictures of the other 2 ferrets in which we visualise IL-6mRNA in endothelial cells in Supplementary Figure 6. We feel that this is an important addition to the manuscript, as it confirms parts our *in vitro* findings in an *in vivo* model.

9. Line 291, "replication kinetics of all three IAV strains changed", please be more specific about the changes authors are alluding to here.

We specified in more detail the changes "*Interestingly, here we show that when epithelial cells were co-cultured with endothelial cells the replication kinetics of all three IAV strains appeared to be slightly faster, and the levels of pro-inflammatory cytokines were increased compared to epithelial mono-cultures.*" (Line 230-233)

10. Please provide more specific information regarding the three different IAV studied here. Why do the authors think that the pdm09 virus infected LMECs to higher levels than the 2019 H1 or H3 viruses in this study (lines89-92)? What molecular changes are present between all three viruses that could assist interpretation of why the pdm09 virus? This data could be easily added to Table 1 with information presented from all three strains employed in the study.

We agree with the reviewer that understanding differences in the viral factors among the studied viruses might explain at least partly the differences contributing to different infection efficiencies. The 2019 H1N1 and H3N2 viruses are genetically very different from each other whereas the 2009 and 2019 H1N1 are genetically more closely related. Understanding which viral factors contribute to these differences would require more in-depth analysis using reassortant viruses. However, this is not the scope of the study and is therefore not included in the manuscript.

11. Lines 404-405, what were the electrical resistance values of cultures at the time of viral infection, and did these values change between mono-culture and co-culture conditions?

We only measured the TEER upon virus infection in epithelial single cultures to understand if viruses can diffuse after breakdown of the trans-epithelial-endothelial barrier through the transwell. However, in the both the epithelial mono- and co-cultures we observed a disruption of the tight junction marker Zona-Occludens 1 which is at least in part responsible for the breakdown of the TEER.

Reviewer #2 (Comments to the Authors (Required)):

This manuscript by Bauer and Rijsbergen uses an exciting model system combining differentiated airway epithelial cells and endothelial cells. As expected they find poor infection of seasonal of endothelial cells alone or in coculture. However they find reduced replication levels in epithelial cells co-cultured compared to epithelial cells alone. Intriguingly they find differences in pro inflammatory cytokines and IFNs in coculture, better mimicking what happens in vivo. Overall then study is well designed and written and provides a strong platform to study epithelial cell-endothelial cell cross talk. The study could be improved by determining if IL-1b, IL-6, or IFNs are responsible for the differences in virus replication between single and co culture, although these experiments would likely be outside the scope of this manuscript. My only minor concern is that figure S3 should be moved into the main text (as a part of Fig 2 or 3) and would be improved by measuring total virus across the time course (area under the curve) and applying statistics to see if the viral load results are significant.

We thank the reviewer for their comment. The data represented in Supplementary Figure S3 are derived from the same experiments as displayed in figure 2A and B. In Figure S3 we only plotted the data of the apical washed of infected mono-cultures and co-cultures per virus in a graph. Since we included all data now in Figure 2, we decided to keep Figure S3 in the supplement. According to the reviewers suggestion, we applied statistical calculations to determine significance of the replication kinetics described this in the manuscript (Line 135-136 and Figure S3).

We agree with the reviewer that determining if IL-1b, IL-6, or IFNs are responsible for the differences in virus replication would be interesting, but as reviewer already suggests these experiments are not the scope of this manuscript.

Reviewer #3 (Comments to the Authors (Required)):

The authors investigate the role of endothelial cells in the pathology of influenza virus and make an important contribution showing that while not all IAV can replicate in endothelial cells the abortive infection and/or the infection of nearby epithelial cells can turn the endothelial cells into producers of cytokine that could contribute to the disease pathology. The figures are very well made and the results are of high scientific quality.

We had a struggle in going through the result section as some results are coming back in different figures. Other results are shown but are not discussed in the discussion so it is unclear if these contribute to the paper. I believe the authors try to tell a chronological story, but perhaps better to separate the results on abortive IAV infection in one part and results on the cytokine secretion in a second part.

So my major comment is that the results part is badly structured and therefore also all the text is a struggle to get through. Maybe some results can be left out so there is no need to discuss them and a better focus is maintained.

1. In fig1 the authors show infection of endothelial cells that are at the basal or apical site of the membrane. The results are identical, which is to be expected. I don't understand why this experiment is in the paper. Later in the study the endothelial cells are always used at the basal side, so what is the value of the apical site grown endothelial cell infection experiment? As it is not discussed I believe this can be omitted from the paper.

In order to confirm that infection of endothelial cells is equally efficient when cells are directly exposed to virus, or when there is a membrane in between the virus inoculum and the cells we performed these experiments. The text has been adjusted accordingly
"To confirm that infection of endothelial cells is equally efficient when cells are directly exposed to virus, or when there is a membrane in between the virus inoculum and LMECs, we cultured LMECs either at the apical or basolateral side of a filter in a transwell system. In LMECs cultured at the apical or basolateral side no increase of infectious virus titers was detected in the apical or basolateral compartments (Fig1A, B)."
(Line 97-99)

2. The results in Fig 2E are already included in Fig 4, or what is the difference? Perhaps better to bring all cytokine profiling together

The data in Figure 2E and Figure 4 are derived from different compartments of the epithelial mono-cultures and epithelial-endothelial co-cultures. In Figure 2E we measured cytokines in the apical compartment and in Figure 4 we measured cytokines in the basolateral compartments. To avoid any confusions, we clarified these differences more clearly in the figure legends as well as in the figures themselves.

3. In fig 1 there is infection/replication of endothelial cell culture, but this also comes back in fig 3, can this not be combined?

Figure 1 shows the direct exposure of endothelial cells mono-cultures to influenza A virus infection. In Figure 2 we show the differences in the infection efficiency between the epithelial mono-cultures compared to epithelial-endothelial co-cultures. For the flow of the results, we feel that this direct comparison of influenza A virus infection in mono-cultures or in epithelial-endothelial co-cultures should be displayed in a separate figure.

4. Title of Fig2 is "Influenza A virus infection and cytokine profiling in epithelial cell cultures alone or in co-culture with endothelial cells." but I don't see the result on IAV replication in epithelial cells alone? The result on cytokine expression of infected epithelial cells is in Fig4, but are there also results on IAV replication in epithelial cells? To avoid any confusion about cytokine profiling of apical and basolateral compartments, we changed the title of figure 2 accordingly "*Influenza A virus infection and cytokine profiling of the apical compartments of epithelial cell single-cultures or in co-culture with endothelial cells.*" (Line 645-647) Furthermore, we added labels into the figure legends so that the reader can easier identify from which compartment the samples were taken for measurements.

5. Perhaps it would be helpful to bring all results related to IAV replication in one part and the results related to cytokine expression in a second part.

As this manuscript contains a lot of data on virus infection, replication and cytokine production we decided to present the data per cell type.

6. In the conclusion the authors write "Interestingly, here we show that when epithelial cells were co-cultured with endothelial cells the replication kinetics of all three IAV strains changed," but we do not have info on replication in epithelial cells in the absence of endothelial cells. Or are the authors referring solely to replication in endothelial cells? There are several occasions where it is difficult to understand what is exactly meant. Please be concise.

To avoid any confusion we focused on viral growth kinetics in epithelial mono-cultures compared to co-cultures and applied statistic to strengthen our argument see figure Suplemnt 3. Furthermore, we specified in more detail that we are focusing on replication in epithelial mono-cultures versus co-cultures. "*Comparing the virus replication in mono-epithelial cultures compared to co-cultures showed that replication was slightly faster in co-cultures (Figure S3).*" (Line 135-136).

7. when vRNA is determined, please always indicate if this is from cells or from cell culture medium, also in the figure legends.

We added the necessary information to the text and figure legends. (Line 104-108, 110-113)

8. In the intro the authors mention the "multibasic cleavage site". please explain and how it relates to this work

As this study focuses on the role of endothelial cells in the pathogenesis of influenza, it is essential to discuss the different observations between highly pathogenic avian influenza viruses (that contain a multibasic cleavage site) with seasonal and pandemic influenza viruses, which do not contain a multibasic cleavage site. To clarify this we have changed the text in Line 70-72.

9. intro: "Suppression of early innate immune responses in endothelial cells and associated cytokine production greatly reduced the mortality of experimentally infected animals." I don't understand how immune suppression can reduce mortality, this seems a contradiction.

We understand the confusion here and explained it better in the text "*It has been shown that suppression of early innate immune responses in endothelial cells reduces the levels of systemic cytokines and decreases tissue damage, both mechanisms contributing to a greatly reduced mortality of experimentally infected animals (10)*"(Line 64-67). The immune response is essential to control an infection, but when this is too strong it contributes to an increase in the disease. Therefore, suppression of early innate immune responses and preventing cytokine storm can result in less severe disease.

10 Intro, something wrong with sentence "For example, high levels of pro-inflammatory responses detected in the blood, are associated with high morbidity in pandemic H1N1 viruses, either from 1918 or 2009. and zoonotic IAV infections in humans, or in nonhuman primates"

We changed the sentence accordingly: "*The severity of IAV infections are dependent on many factors such as virus replication as well as the hosts immune response (1,3,4). For example, a dysregulation of the pro-inflammatory cytokine response in the lung and high levels of proinflammatory cytokines in the blood of infected individuals, are associated with high morbidity caused by pandemic H1N1 viruses, either from 1918 or 2009 or zoonotic IAV infections.*" (Line 52-57)

11. When measuring cytokines it is not always indicated if this is RNA expression or protein that is measured. Please always indicate where needed.

We clarified throughout the text if either gene expression or protein concentrations were measured.

line 107: after IFN-L treatment there is IFIT1 induction, so why does this suggest that IFNLR is not expressed? This is also not discussed are related to later in the paper. Is this then not relevant? Or I missed it perhaps.

As also mentioned by reviewer 1 in Question 1, we have changed the text accordingly (Line 120-124).

line 108: "we do not detect infectious virus progeny in the supernatant" where is this shown?

To avoid any confusion we added the reference to figure 1A and 1B to this statement (Line 124)

line 110: "The intracellular genome replication 110 induces a type-I and type-III interferon response." - better to move this to line 104

We moved the sentence up to Line 104

general: I don't like "single cultures" perhaps better "mono cultures" as this better compares with co-cultures. I'm not sure if you always have to indicate that it is a mono culture, this is clear from context I believe.

We changed single cultures to mono-cultures throughout the text.

abbreviation "TER", I believe the rest of the world uses TEER?

We agree with the reviewer and corrected TER to TEER in both the text and supplementary figure. (Line 139)

March 30, 2023

RE: Life Science Alliance Manuscript #LSA-2022-01837R

Dr. Debby van Riel
Erasmus Medical Center
Netherlands

Dear Dr. van Riel,

Thank you for submitting your revised manuscript entitled "The pro-inflammatory response to influenza A virus infection is fueled by endothelial cells". We would be happy to publish your paper in Life Science Alliance pending final revisions necessary to meet our formatting guidelines.

- please address the final Reviewer 3's minor points
- please add ORCID ID for first corresponding author-you should have received instructions on how to do so
- please add the Twitter handle of your host institute/organization as well as your own or/and one of the authors in our system
- please provide an ethics statement for the patient from which H1N1 virus was collected

A. FINAL FILES:

B. MANUSCRIPT ORGANIZATION AND FORMATTING:

Sincerely,

Reviewer #1 (Comments to the Authors (Required)):

Authors have satisfactorily addressed all comments raised during initial peer review; no additional comments.

Reviewer #3 (Comments to the Authors (Required)):

Thank you for adapting the manuscript. For me it greatly improved the reading. And I agree with all the points in the rebuttal of the authors. The structure of the paper is now more evident to me and separating the results base on cell-model (mono- or co-) is indeed more logical. But as I now understand the setup better I have some additional comments.

One important point relates to the staining of ferret and human lung tissue. The authors may want to add in the conclusions something like: "From these analysis we cannot conclude on a potential up-regulation of IL-6 mRNA expression due to IAV infection as we have not explored tissue of uninfected animals and patients." This is important as there now seems to be an assumption that in these ex vivo / in vivo experiments there is an up-regulation due to infection while the uninfected controls are clearly lacking.

A less important point relates to the bar-charts in fig 4. There are several points with "induction over mock = 1". I believe this is an error in data representation. Perhaps the Y-axis should start at 0.1 or 0.001. There might be a detection limit, this can than be indicated as a horizontal dotted line. Obviously the detection limit must be lower than 1. I realize these are less important data points but showing them as "1" makes it difficult to understand. Or it can be added to the figure legends that induction of <1 is shown as 1.

Some minor points:

line 67: "both,"  ", both"

line 119: "Intracellular genome replication ... response." It is unclear if this is a conclusion of an experiment or it is a reference to the literature. Perhaps the authors want to say " We also investigated if ..."

line 134: put a "." at the end of the sentence

line 136: "as co-cultures measured"  "as in the co-cultures when measured"

line 156: there is detection of IL-8 but this is not shown in Fig2F, while IL-1b is shown in the fig but not mentioned in the text.

line 182: "No large differences in gene expression ... inoculation."  is this in AO or LMECs? or both?

line 197: IL-8 is above the detection level. Why is this mentioned? Is there an induction by IAV or no change?

line 245: "antiviral state"  "an antiviral state"

line 245: "studies show"  "studies on endothelial cells show"

line 260: TER  TEER

line 651: co-cultured  co-culture

Reviewer #3 (Comments to the Authors (Required)):

Thank you for adapting the manuscript. For me it greatly improved the reading. And I agree with all the points in the rebuttal of the authors. The structure of the paper is now more evident to me and separating the results base on cell-model (mono- or co-) is indeed more logical. But as I now understand the setup better, I have some additional comments.

One important point relates to the staining of ferret and human lung tissue. The authors may want to add in the conclusions something like: "From this analysis we cannot conclude on a potential up-regulation of IL-6 mRNA expression due to IAV infection as we have not explored tissue of uninfected animals and patients." This is important as there now seems to be an assumption that in these ex vivo / in vivo experiments there is an up-regulation due to infection while the uninfected controls are clearly lacking.

We have included the following sentences to the results section to make clear that negative control tissues were not included in these analysis (Line 210-215): "From this analysis we cannot formerly conclude that the influenza A virus infection was directly associated with the expression of IL-6 mRNA in endothelial cells as no uninfected tissue were included".

A less important point relates to the bar-charts in fig 4. There are several points with "induction over mock = 1". I believe this is an error in data representation. Perhaps the Y-axis should start at 0.1 or 0.001. There might be a detection limit, this can than be indicated as a horizontal dotted line. Obviously the detection limit must be lower than 1. I realize these are less important data points but showing them as "1" makes it difficult to understand. Or it can be added to the figure legends that induction of <1 is shown as 1.

We have addressed this by adding the following text to the figure legend 4 (Line 693-694): "If the induction of cytokines in any condition was <1, it is shown as value 1."

Some minor points:

line 67: "both,"  ", both"

This has been adjusted (Line 66).

line 119: "Intracellular genome replication ... response." It is unclear if this is a conclusion of an experiment or it is a reference to the literature. Perhaps the authors want to say " We also investigated if ..."

This has been adjusted (Line 117).

line 134: put a "." at the end of the sentence

This has been adjusted (Line 133).

line 136: "as co-cultures measured"  "as in the co-cultures when measured"

This has been adjusted (Line 135).

line 156: there is detection of IL-8 but this is not shown in Fig2F, while IL-1b is shown in the fig but not mentioned in the text.

This has been adjusted (Line 155).

line 182: "No large differences in gene expression ... inoculation."  is this in AO or LMECs? or both?

We have changed the text as follow to make this more clear (Line 180-182): No large differences in the upregulation of *IFIT1* or *IFN-β* gene expression were observed following pH1N1 virus infection when comparing lysates of apically infected AO at ALI or LMECs

line 197: IL-8 is above the detection level. Why is this mentioned? Is there an induction by IAV or no change?

This has been removed.

line 245: "antiviral state"  "an antiviral state"

This has been adjusted (Line 243).

line 245: "studies show"  "studies on endothelial cells show"

This has been adjusted (Line 244).

line 260: TER  TEER

This has been adjusted (Line 259).

line 651: co-cultured  co-culture

This has been adjusted (Line 653).

April 5, 2023

RE: Life Science Alliance Manuscript #LSA-2022-01837RR

Dr. Debby van Riel
Erasmus MC
Netherlands

Dear Dr. van Riel,

Thank you for submitting your Research Article entitled "The pro-inflammatory response to influenza A virus infection is fueled by endothelial cells". It is a pleasure to let you know that your manuscript is now accepted for publication in Life Science Alliance. Congratulations on this interesting work.

DISTRIBUTION OF MATERIALS:

Again, congratulations on a very nice paper. I hope you found the review process to be constructive and are pleased with how the manuscript was handled editorially. We look forward to future exciting submissions from your lab.

Sincerely,
